# Automated vitrification of cryo-EM samples with controllable sample thickness using suction and real-time optical inspection

Roman I. Koning [1✉], Hildo Vader[2], Martijn van Nugteren [2], Peter A. Grocutt[2], Wen Yang[3], Ludovic L. R. Renault [3], Abraham J. Koster [1], Arnold C. F. Kamp[2] & Michael Schwertner [2]

The speed and efficiency of data collection and image processing in cryo-electron microscopy have increased over the last decade. However, cryo specimen preparation techniques have lagged and faster, more reproducible specimen preparation devices are needed. Here, we present a vitrification device with highly automated sample handling, requiring only limited user interaction. Moreover, the device allows inspection of thin films using light microscopy, since the excess liquid is removed through suction by tubes, not blotting paper. In combination with dew-point control, this enables thin film preparation in a controlled and reproducible manner. The advantage is that the quality of the prepared cryo specimen is characterized before electron microscopy data acquisition. The practicality and performance of the device are illustrated with experimental results obtained by vitrification of protein suspensions, lipid vesicles, bacterial and human cells, followed by imaged using single particle analysis, cryo-electron tomography, and cryo correlated light and electron microscopy.

[1] Electron Microscopy, Cell and Chemical Biology, Leiden University Medical Center, P.O. Box 9600, 2300 RC Leiden, The Netherlands. [2] Linkam Scientific Instruments Ltd, Tadworth, Surrey KT20 5LR, UK. [3] NeCEN, Institute of Biology Leiden, Leiden University, Gorlaeus Building, Einsteinweg 55, 2333 CC Leiden, The Netherlands. ✉email: r.i.koning@lumc.nl

Cryo fixation into vitreous water (amorphous ice) by fast freezing of biological samples can deliver nearly perfect structural preservation of biological samples like protein suspensions, viruses, bacteria, and eukaryotic cells. Cryo fixation requires a freezing rate high enough (>100.000 °C/s) such that ice (crystal) formation is prevented. As a result, the water adopts a glass-like amorphous, metastable transient state[1]. Using vitrification, the structure of proteins and cells can be preserved in their native hydrated environment to atomic resolution. Vitrified samples are compatible with the vacuum conditions required for cryo-electron microscopy (cryo-EM), and can also be studied with optical cryo-fluorescence light microscopy (cryofLM)[2]. Correlative light and electron microscopy (CLEM)[3] brings together the advantages of EM (high resolution, structural context) with the advantages of the wide range of available light microscopy techniques (live imaging, versatile labelling)[4,5].

Vitrification via plunge-freezing using liquid ethane, or an ethane/propane mixture as a cryogenic[6], was shown to be a practical approach for cryo preparation of biological samples up to 10 microns thick[1,7]. For cryo-EM, purified protein and virus suspensions are preserved in thin water layers measuring several tens of nanometres, from which atomic resolution reconstructions can be determined using SPA[8,9]. Larger structures, such as bacteria and adherent cells up to a few microns in thickness, are also suitable for vitrification. Three-dimensional reconstructions with molecular resolution can be determined using cryo-electron tomography (cryo-ET) of samples up to approximately half a micron thickness[10,11]. Minimizing the thickness of the liquid layer is important since medium surrounding the sample scatters electrons, adding to background noise in the images, thereby lowers the signal-to-noise ratio in the images and reducing the attainable resolution in resulting three-dimensional reconstructions.

The essential step of generating a thin liquid sample layer on an electron microscopy specimen support for EM (typically a holey carbon layer supported by a 3.05 mm diameter copper grid) is problematic, as thin water layers are inherently unstable and to have exact control over the water layer thickness is difficult. It was found that rendering the support film hydrophilic by glow-discharging in air or alkylamine[12,13] helps to form a thin liquid layer on the support film and a humid-saturated environment helps to stabilize the thin layer. Current common practice is to apply several microliters of specimen solution to a glow-discharged support film, followed by blotting away excess fluid using filter paper which is subsequently plunge-frozen[14,15].

However, this method has several challenges. First, it is difficult to control the water layer thickness and determine its distribution over the grid. Because the water layer thickness is influenced by many factors, including properties of the support, the sample (type, concentration, buffer, solutes, etc.), and the environmental conditions (temperature and humidity)[16], optimal parameters are often found via time-consuming trial and error involving iterative cycles of freezing and analysis by cryo-EM. Second, relatively large amounts of sample are lost during the process, since most sample is absorbed by the blotting paper, resulting in less than one permille of the sample remaining on the grid[17]. Third, it is reported that the use of blotting paper can result in disadvantageous effects, like protein aggregation and denaturation[18,19]. Furthermore, manual sample application, handling of tweezers, and transfer between containers and different machines are time-consuming and require significant training and user skills. Because current state-of-the-art applications of cryo-EM features robotic sample loading of multiple grids, highly automated data recording[20,21], and on-the-fly image processing[22], the speed and quality of sample preparation have become bottlenecks in the overall process.

Alternative sample preparation methods to blotting have been proposed and developed, which include the application of minimal amounts of a sample using spraying[23,24], inkjet dispensing[25,26], microcapillary writing[18], or contact pin printing[27]. These methods do not need to remove excess liquid off the grid and precious sample is used efficiently. Also, the recently developed vitrification systems are highly automated and minimize steps related to handling, repeatability, and speed problems. Some drawbacks of these systems are that sample spreading over the grid is spatially limited or that specialized grids are necessary for good spreading[28]. In addition, most systems are suitable for the vitrification of suspensions but not specifically designed to work for samples that are much larger, such as adherent cells or bacteria. Moreover, none of these methods address resolving time-consuming test cycles between vitrification and quality assessment by cryo-EM[29].

Here we set out to develop a vitrification device for cryo microscopy that (i) allows determining the usability of a vitrified specimen before performing cryo microscopy, (ii) is compatible with all types of samples (proteins, bacteria, and cells), (iii) produces grids in a reproducibly way with controllable sample ice thickness, and (iv) has a high degree of automation. We present a plunge-freezing device that has extensive automation of the sample preparation process, including grid handling, glow discharging, control of cryogenic liquids, and sample application. The device uses sample removal by suction—not filter paper—and allows visual inspection of the grid during thin film formation. The observation of interference patterns by light microscopy, in combination with dew-point temperature control of the grid, enables precise control of the water layer thickness and determination of the optimal moment for vitrification. Optical monitoring of thin-film formation before plunging proved to be a usable and reliable method for quality assessment quality before time-consuming cryo-EM analysis. The results that we report here show that the device can be used for vitrification of proteins, liposomes, viruses, bacteria, and cells for single-particle, tomographic, and CLEM cryo-electron microscopy techniques.

## Results

**Plunging workflow.** The Linkam plunger is a highly automated device that prepares samples on an EM support grid for cryo transmission electron microscopy. Key features are that sample application is performed by grid immersion in the sample solution, sample film thickness adjustment is performed by slowly retrieving the grid from the sample solution followed by suction. The air dewpoint temperature of the grid is controlled, and the formation of the water layer thickness on the grid is inspected (and recorded) live by transmission and/or reflection light microscopy. The workflow of this Linkam system differs from other grid plunging workflows since the subsequent steps of grid retrieval from a storage box, glow discharging, sample application, liquid removal, vitrification (sample plunging into a cryogenic solution), and loading into a cryogen storage box are performed automatically (Fig. 1). In other plunge-freezing devices, i.e., the Thermo Fisher Scientific Vitrobot and the Leica EM GP, the most important step is situated between blotting and vitrification (which determines specimen thickness) and is determined by a pre-set time parameter. In the Linkam plunger, all steps are automated, except for the selection of the specimen thickness, allowing precise timing of vitrification.

**Detailed plunger description.** The design of the cryo-section of the Linkam plunger (Fig. 2) is related to that of the Linkam CMS 196 cryo stage[30]. The plunger contains a (1) liquid nitrogen storage container (Fig. 2 a left); (2) a central (covered) region

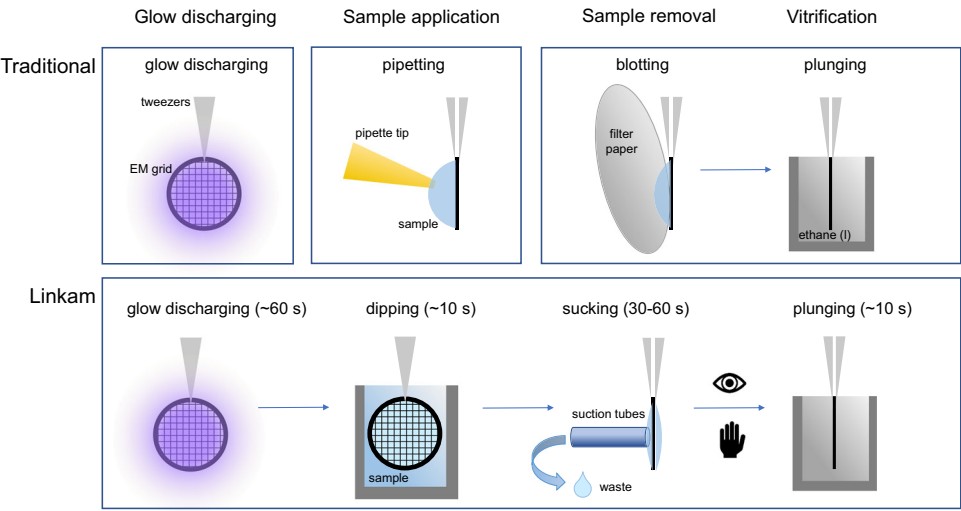

**Fig. 1 Workflow comparison.** Schematic overview of the workflow of conventional plunging devices (top) where glow discharging and sample application are not integrated with automated sample removal and subsequent plunging. In the Linkam approach (bottom) all steps are integrated into an automated workflow. In this approach, the sample application is not performed by pipetting but by dipping the grid in solution and sample removal is not performed by blotting with filter paper but by suction with a tube. The process of thin layer formation is followed live and only the timing of plunging is done manually, exactly opposite to the traditional workflow.

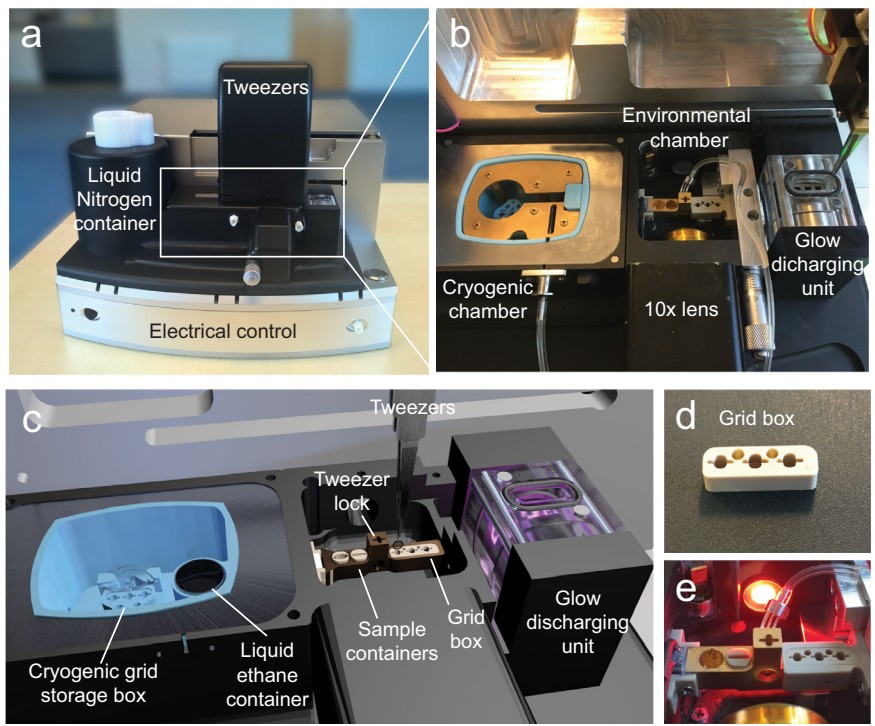

**Fig. 2 Design and functionality. a** Linkam plunger layout with liquid nitrogen container (left), tweezers control (top) around the central chambers (white box). **b** Central chambers with a cryogenic chamber (left), environmental chamber (middle) and glow discharger (right), and lens (bottom). **c** The cryogenic chamber is filled with liquid nitrogen from the liquid nitrogen container (see **a**) and contains a cryogenic grid storage box and a liquid ethane container (still from Supplementary Movie 1, for clarity the ethane gas tubing and brass cover are not shown), the central environmental chamber contains two positions for sample containers and a place for three grids. Centrally there is a temperature-controlled 'tweezer lock' that cools the tweezer while it puts the grid in position for water removal by suction pipes (not shown) and simultaneous visual inspection by the lens. **d** Grid box for three grids. **e** Temperature-controlled block with sample containers and lock for tweezers and slit for grid positioning for suction tubes and light microscopy.

containing the cryogenic container, environmental chamber, and glow discharging unit (Fig. 2b, c); (3) a digital camera and a ×10 light microscopy lens (resp. behind and in front of the containers); (4) mechanically controlled and movable tweezers (over the chambers), which all are mounted on (5) an aluminium frame containing the electrical units and pumps. The Linkam plunger and the camera are connected to a laptop with control software (not in figure).

The liquid nitrogen Dewar (Fig. 2a) contains up to 200 ml liquid nitrogen which can be filled through a funnel and shielded with a lid with an open insulating tube for nitrogen gas outflow (not present in Fig. 2). The liquid nitrogen inside the Dewar is

monitored by a temperature sensor and the level in the cryogenic chamber is kept constant with a valve controlling the top-up.

The cryogenic chamber (Fig. 2b, c) contains a location to fit a custom-made cryogenic grid storage box, which can hold three vitrified grids for transport and storage and the liquid ethane container. The ethane bath and associated gas filling tubes (not shown) are temperature controlled to allow automated condensation of ethane gas. Constant level and temperature (at −183 °C) are maintained for the liquid cryogen bath to prevent the freezing of ethane. Since the cryogenic chamber is relatively small the top of the cryogenic chamber is shielded by a cooled copper plate (Fig. 2b) to maintain the cryogenic temperature of the gas phase.

The environmental chamber (Fig. 2b, c) is temperature-controlled by a Peltier element and can be set between 3 and 50 °C. The bottom of the chamber can contain a few mL of demineralized water to generate humidity by evaporation. The grid area contains a removable grid box (Fig. 2c, d) that can store up to three EM grids that can be picked up and used for glow-discharging and subsequent sample application. Alternatively, it contains a box with adherently grown cells on EM grids in liquid. The environmental chamber also contains two positions for liquid sample application containers. A ×10 lens integrated microscope setup with LED transmitted and reflected light modes can image the EM grid inside the humidity chamber while the tweezer holding the grid is inserted into the 'tweezer lock' in front of the lens. This temperature-controlled brass 'tweezer lock' (Fig. 2e) contains a slit through which the tweezers with the grid can be positioned down into a hollow cylinder. Here the grid is positioned for simultaneous sample removal via suction and light microscopy imaging. An interchangeable suction module with up to three suction tubes is positioned against the outside rim of the grid for the removal of excess fluid. The temperature of the "tweezer lock" is set at a lower temperature than that of the chamber such that the tweezer tip and the grid can be set around the dewpoint to control water evaporation and thereby water layer thickness and simultaneously prevent a change in sample concentration.

The glow discharging unit has three positions for grids which are located between two electrodes. The volume is automatically pumped to $1.8 \times 10^{-1}$ mbar within a minute using a small oil-free vacuum pump. Glow discharging is automatically performed according to programmable settings, typically within 2 min at 5 mA in air.

**Practical workflow**. Preparation of the Linkam plunger for use is done by powering on all electrical units (Peltier water cooler, lens heater, plunger, and support PC), connecting or filling all consumables (ethane gas cylinder, liquid nitrogen in the Dewar, water in the environmental chamber, grid storage box in the cryogenic container, EM grids into the grid pickup cartridge, liquid sample in the sample container) and starting the plunger and camera control software and starting the liquid ethane filling protocol. A detailed protocol is presented in Supplementary Note 1.

Cryo-grid preparation is performed automatically after starting the programmed sequence (Supplementary Movie 1). First, a grid is picked up by the tweezers and transferred into the glow discharge unit where it is released and rendered hydrophilic in air plasma. Next, the glow discharged grid is again picked up and transferred into the sample-filled container where the sample is applied to the grid by dipping into the temperature-controlled sample liquid. A minimum of 10 μL is needed to fill the sample container. By dipping <0.5 μL is actually applied on the grid, which is <2–3 μL that is generally used by manual application. Grid soaking time, as well as the retraction speed of the grid from

the liquid container, are both programmable. The grid is subsequently placed in front of a temperature-controlled ×10 light microscopy lens by the tweezers. The tweezer tip and the grid held in it are both temperature-controlled in this configuration because the tweezer is touching the brass temperature-controlled "tweezer lock", set a few degrees under the dewpoint of the temperature of the environmental chamber. Then, a suction module containing two or three tubes is placed in front of the grid, touching the rim of the grid, and upon activation excess fluid is sucked away with an adjustable and calibrated flow. Simultaneously, the thickness of the water layer on the grid is visually monitored by the microscope with the attached camera and the timing of transfer to the liquid ethane container and vitrification by plunging is triggered by the operator based on the real-time camera image. After vitrification, the grid is transferred into the cryogenic grid storage box or a cryo-cassette for cryo-fluorescence imaging in the CMS196V3 cryo-stage.

**Determining the moment of vitrification**. To provide a view on the grid for real-time light microscopy observation, tubes were used for to removal of excess fluid from the EM grid instead of filter paper (Figs. 3 and 4, Supplementary Movie 3). Light microscopic observation of 'bulk liquid' on the grid following application of sample by dipping does not result in any specific feature that leads to a thickness estimation of the water layer (Fig. 3a). When the water thins, coloured thin-film interference patterns start to emerge, generated by the interference of light emanating from the air–water interface on the carbon side of the grid and the carbon film (Fig. 3b). Upon further removal of fluid these coloured interference fringes disappear, when the distance between the water surface and carbon layer becomes less than ~200 nm (Fig. 3c). After further fluid removal, a cross-like light pattern appears when the meniscus reservoir at the side of the grid is formed (Fig. 3d). At that moment a bright disc appears in the centre of the square (Fig. 3e) because the water layer thickness in the centre yields is reduced to roughly less than a micron (as observed from the resulting electron transparency after vitrification). On the sides, a thicker water meniscus is still attached to the grid bars—appearing in a dark rim (Fig. 3f). Upon further water removal this water might also be completely removed (not shown here), but in our experience then also the centre of the grid is dried out. Cryo-EM observations showed that only the latter two 'states' have, after vitrification, an electron transparent vitrified water layer (Fig. 3e and f), while the other states exceed the penetration depth of electrons, resulting in black squares in the cryo-EM images. So, the optimal moment for vitrification is when grid squares, when observed by light microscopy, appear to have a bright central region and a dark outer ring. To obtain a large field of view in our experiments we mainly used a ×10 objective lens. However, using a ×20 lens allowed to more clearly observe differences between open and water-covered holes (using Quantifoil 2/2 carbon foil), which allowed for more precise determination of the point of vitrification, at the expense of knowledge on the thickness variation of the water layer over the grid (Supplementary Fig. 3C).

**Controlling water layer thickness variation over the grid**. Initial experiments showed that sucking away fluid using a single suction tube positioned on the bottom side of the grid, resulted in a steep thickness gradient between water-filled squares and dry squares. As a consequence, at any time during the process, only a very limited number of squares showed the desired thickness that was suitable for data collection. To obtain more squares with optimal thickness, and a better distribution over the grid, we tested several layouts of suction tubes. It appeared that

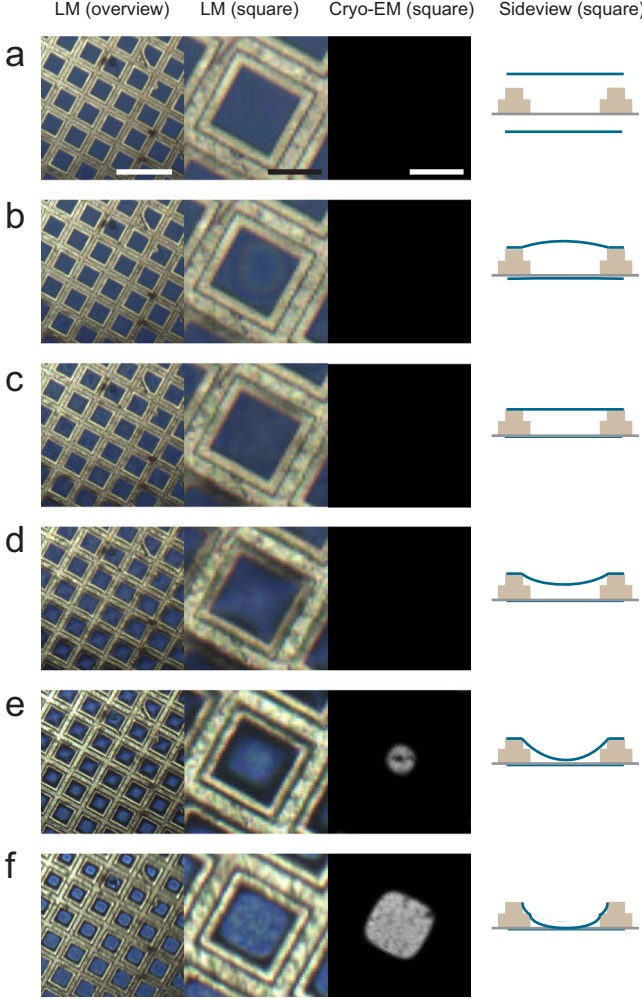

**Fig. 3 Microscopic view on thin film formation.** Assessment of film thickness by real-time LM and cryo-EM. First two columns: LM observations (grid overview, grid square), third column: cryo-EM view of the same square after vitrification, last column: schematic side view explanatory figure of one square with grid bars in lilght brown, carbon layer in grey and liquid surfaces in blue lines. **a** Just after sample application by dipping, the bulk sample solution is present on both sides of the grid and no specific features can be observed by light microscopy. The schematic overview shows the liquid surface (blue lines), the grid bars (beige), and the carbon layer on one side of the grid (grey). **b** After the removal of water, coloured interference rings appear, presumably due to light interference between the surface of the water and the carbon layer on the carbon side of the grid. Interference lines can appear within one square but can also span multiple squares. **c** Interference patterns disappear and near grid edges the appearance slightly changes due to touching of the water layer with the grid bars. **d** In the centre of the grid squares a lighter spot appears, presumably because the water surface adopts a concave shape within the square. **e** Again, interference rings are emerging, now due to interference between the other water layer in and the carbon layer (see Supplementary Movie 3). The lighter spot represents water layers that are thinner than a few hundred nanometres: cryo-EM images this square shows electron transparent central regions. Darker regions on the edges of the LM image are probably the result of light rays being refracted towards and absorbed by the grid bars. **f** The central region expands in the LM image. Cryo-EM shows an expansion of the electron transparent region while a soft edge appears on the edges. Views like in **e** and **f** are the right timing for vitrification and result in usable grid squares for cryo-EM imaging. A good and simple visual criterion is that the black frame around the grid square recedes again after surpassing the largest area as shown in (**e**).

positioning two suction tubes on the left and right sides of the grid gave the most reproducible and the best results (Supplementary Fig. 2). Appearance, quality, and overall usability for data collection of the produced grids (Fig. 4) were comparable to those vitrified in our lab using the Thermo Fisher Scientific Vitrobot or Leica EM GP.

Since water removal is done from the sides of the grid while at the (central) position of the tweezer tip water is often retained on the grid, a thickness gradient is created, which enables proper timing of vitrification to produce a large area of usable grid squares. It must be said that the sample itself (composition, viscosity, surface tension) influences how this gradient behaves and how much water is retained on the grid. Also, the type of grids (mesh, vendor, thickness) and the quality (wrinkles, tears, holes) of the carbon layer as well as carbon film type (lacey or quantifoil: amount and size of holes), influences water behaviour on the grid during layer thinning by suction. Individual differences between grids and samples however did not influence the quality of the vitrified grids. During preparation every sample is individually assessed.

Furthermore, LM observation allowed observation of several phenomena that proved useful to assess the quality of the grid. First, aggregation of proteins and vesicles on the grid could be observed during water removal by suction. Aggregates in the microns size range lead to local thickness variations of the water surface that translate into contrast variations that can be observed by LM. Aggregation of protein within the holes in the foil could not be observed in the current setup. Also, hydrophobic patches on the carbon foil were occasionally observed by non-uniform retraction of the water edges (Supplementary Fig. 3).

**Controlling ice layer thickness**. Although visual inspection of grid squares in the optical image before plunging is a useful indication for the quality and usability of the resulting vitrified grid by cryo-EM, the ice film thickness over the holes in the carbon layer is of key importance. The individual two micrometre diameter holes in the Quantifoil R2/2 EM grids can be observed with a ×10 NA 0.25 lens in combination with a 5 MP digital camera, allowing to investigate whether the holes stay covered with water or pop open by the large surface tension. The general tendency was that in squares with water at its rims, the holes in the carbon film are covered. When little water is present at the outside rims of the square, the holes are open, not covered with water.

While the presence of bulk water is controlled by suction, inherently unstable thin water layers are sensitive to the relative humidity of their environment. Since the instrument has no active humidity control (to increase the environment humidity to reduce evaporation) the grid is surrounded by variable humidity of around 79% (typical in The Netherlands). Therefore, the tweezer tip and thus the grid were cooled to a few degrees under the set temperature in the environmental chamber using a "tweezer lock" (Fig. 2c) to maintain the grid at the dewpoint and avoid evaporation[18]. This way we were able to control the evaporation of water from the grid and maintain favourable states for vitrification (Fig. 3e and f) for several minutes (Supplementary Fig. 4 and Supplementary Movie 2).

**Application results**. To validate the performance of the vitrification device we tested a variety of samples and techniques. Apart from adapting solution concentrations, we did not optimize any parameters and all samples were made within a few vitrification sessions. For cryo-EM single-particle analysis, we vitrified freshly purified apoferritin and prepared a few grids. From one of the grids we recorded several thousand images and determined a

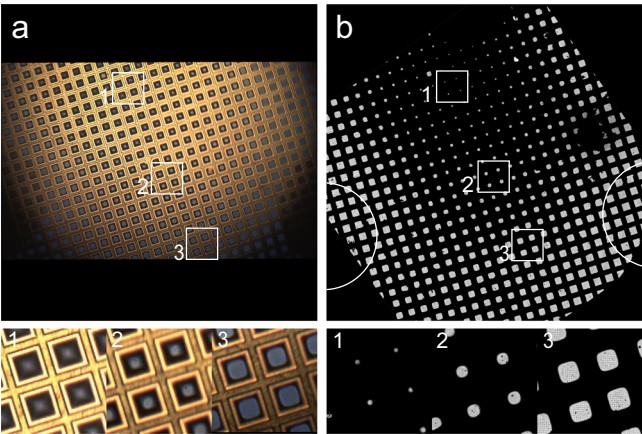

**Fig. 4 View on sample distribution and thickness on the grid by live light and cryo-electron microscopy. a** LM overview of EM grid with liquid sample (top, last movie frame before vitrification) and **b** cryo-EM overview of the grid after vitrification (top). Numbers 1, 2 and 3 in **a** and **b** denote the same grid squares showing the appearance in LM and EM of different sample thicknesses. Optimal sample thickness is achieved around position 2. Note that in **b**, inset 3, most holes on the support film are not covered by an ice layer, and **b**, inset 2, contains more usable holes in the support film for data acquisition. White circles denote positions of suction tubes.

2.4 Å reconstruction into which we could fit the X-ray structure (PDB entry 6RJH) (Fig. 5a).

For Cryo-ET, we vitrified DNA origami cubes. This is a sample that tends to aggregate and stick to the air–water interface. Though aggregation was not significantly reduced, we were able to perform cryo-ET. The tomographic reconstruction clearly showed the DNA as the ribbons of the cube (Fig. 5b). In another experiment, aimed at 2D cryo-EM imaging of liposomal vesicles, amphotericin B vesicles (Ambisome®) were vitrified. Cryo-EM images of these samples showed good distributions of covered holes within a grid square and good images from many holes were obtained (Fig. 5c two left panels). Samples were also prepared cryo-ET on multilamellar vesicles, revealing multiple membrane layers of the vesicles (Fig. 5c two right panels).

In addition to being suitable for protein and liposomal suspensions, the vitrification device also allows for the preparation of cells. This is demonstrated by the vitrification and cryo-ET imaging of both *E. coli* bacteria suspensions and 17 clone 1 mouse cells that were adherently grown on gold EM grids (Fig. 6a, b). Finally, we also performed cryo-CLEM on these eukaryotic cells, fluorescently stained with Mitotracker, by subsequent cryo-LM imaging (Fig. 6c, left) and additional cryo-EM (Fig. 6c, middle, right).

## Discussion

We had two main incentives to develop and build this vitrification device for the preparation of cryo-EM samples. A first goal was to make the vitrification process more consistent, less user-dependent, and easier to control. Therefore, the grid handling steps were automated, from picking up an EM grid (out of the custom grid box; Fig. 1d) to putting a fully processed cryo-EM grid in a cryogenic container for storage (and for cryoLM imaging in specialized cassettes)[30]. By also integrating glow-discharging and sample application, the user interaction with the grid is minimized having the advantage that grid degradation (wrinkling) or loss are minimized. To increase the convenience of usage, liquefaction of cryogenic gas (ethane or ethane/propane) and maintaining the cryogenic liquid (ethane and nitrogen) levels, are fully automated.

A second goal was to prepare cryo-EM grids with a more reproducible thickness of the vitreous water layer and get direct feedback on the sample thickness and distribution on the grid during preparation. Having prior knowledge on the thickness of the vitreous water layer, and the general usability of the prepared cryo sample, before performing the cryo-electron microscopy imaging, prevents time-consuming cycles of specimen preparation and grid quality check in the cryo-electron microscopy. To access thickness layer information during the removal of water, we refrained from the use of filter paper, which obscures the view on the grid, and instead decided to use aspiration to remove the excess liquid, by making use of several tubes that are positioned at the outer rim of the electron microscopy grid. Using suction both speed and duration of liquid removal can easily be regulated, in contrast to conventional filter paper, as the speed of the removal is an intrinsic property of the paper. Our method has the additional advantage that it removes any potential influence of the filter paper on the chemistry of the sample[18,31]. A strong advantage of the current setup using liquid aspiration is that it enabled the positioning of a light microscopy lens with camera and lighting for reflection and transmission modes around the grid and observing the grid during sample removal. Initial experiments using a ×20 lens provided a relatively large field of view on about 45 squares (5 × 9 squares, 423 × 762 μm) with a useable view of the 2-micron holes. The current and preferred setup makes use of a ×10 lens (16 × 26 squares; 1.35 mm × 2.2 mm with an upgraded 20 MP camera) providing a less detailed view of individual 2-micron holes but offers a relatively large view corresponding to about 70% of the total grid area that can be observed in EM.

Comparison of the recorded light microscopy during preparation and the resulting cryo-electron microscopy of the vitrified grids demonstrated that the thickness of the vitrified water in cryo-EM could be qualitatively estimated using light microscopy. From light microscopy images, the areas that are electron lucent in cryo-EM could easily be identified and even the difference between water-filled and empty holes in the carbon could be observed. Even though the time between the last LM observation and vitrification takes up to a second, it is fast enough such that not many changes appear. The resulting thickness of the vitrified water layers was not quantitatively measured but on the basis of the particle size in the different samples and tomography data we estimated that the resulting vitrified water layers vary between around 50 and 200 nm. Altogether, the light microscopy observation of thin film interference gives a good estimation for a suitable timepoint of vitrification and results in grids with a vitrified water layer thickness that are practically usable.

Though white light thin layer interference can give detailed information on film thickness[32], in the current setup with transmission and reflected light, the exact colour cannot be determined. Also, below ~100 nm thin films are colourless and only changes in intensity, which cannot be related to a specific thickness since these are masked by background intensity differences. However, quantitative thin film analysis in the range of 50–200 nm is important for determining optimal thickness for high-resolution cryo-EM of small particles and it would be a desirable feature for a plunger setup. With appropriate spectrally resolved detection and analysis methods the quantitative thickness of the liquid film could be determined optically. In principle, interference colour analysis using light of three wavelengths[33] or a combination of holography and interferometry[34] were demonstrated for full-field optical thin film measurement. Though measuring films below 100 nm thickness will likely need shorter wavelengths.

Bulk water can be efficiently removed from the rim of the grid by suction using two or three tubes. This resulted in a slightly

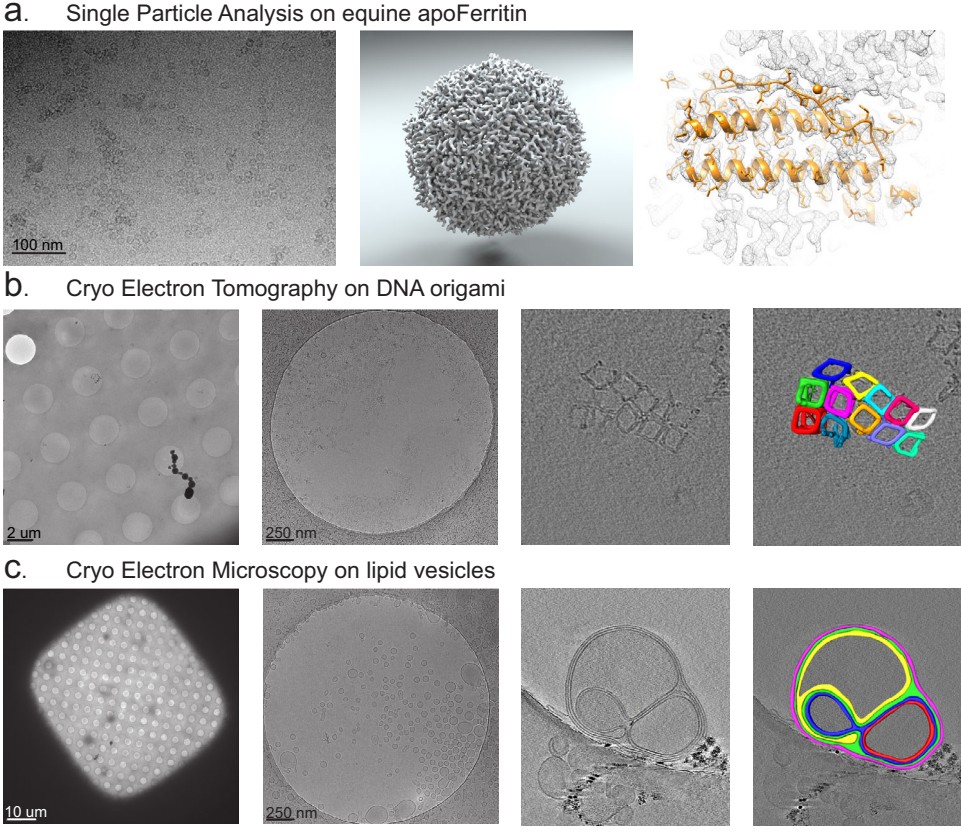

**Fig. 5 Application results, single particle analysis. a** Single particle analysis results for apoferritin ($n = 1$). Typical cryo-EM image from data collection (left), 3D reconstruction of apoferritin at 2.4 Å resolution (middle), and fit of X-ray structure of horse spleen apoferritin (PDB 6rjh in yellow) in the EM density (right). **b** Cryo-ET on DNA origami cubes ($n = 1$). Cryo-EM view of several holes in holey carbon foil (left) and of the distribution of particles within one hole (centre left), a slice through the tomogram volume showing the top of the DNA ribbons (centre right), and a 3D surface rendering of 12 DNA origami cubes (right, individually coloured). **c** Cryo-EM of lipid vesicles ($n = 5$). Cryo-EM overview showing the ice thickness in an EM grid mesh square that is typically used for data collection (left) and an overview of the distribution of vesicles within one hole in carbon film (centre left), a slice through a tomogram volume of a multi-layered vesicle (centre right) and the 3D surface rendering of the individual lipid layers (right, with coloured lipid layers).

varying distributed water layer thickness over the grid, with the thicker areas near the centre of the grid and the tweezer tip, and the thinner areas near the periphery of the grid. This ice thickness variation is desirable and ensures that on every grid a minimum number of usable squares are present. Optimization of the suction speed is possible, but in our hands, this was not necessary.

Since visual inspection gives prior knowledge on the quality and usability of the grids for cryo-EM, only good grids are transferred into the TEM and used for data acquisition and we have a perfect yield of grids, while discarding the occasional grids that did not behave well during thinning, e.g., because of bending or broken support film, before plunging. For purified protein and liposomal samples, within a single grid typically one-third of the squares (especially around the position of the suction tubes) is too dry and unusable for data collection, one-third is usable (vast majority of foil holes are covered with vitreous water) and one-third has roughly equal numbers of empty and filled foil holes, making the usable number of squares per grid vary between 30% and 60% ($n = 16$). For grids with bacteria and adherent cells the yield was much better since water is better retained around these structures and they are less sensitive for drying.

While having prior knowledge of the thickness of vitreous water layers and the distribution and amount of usable surface on a grid before one performs cryo-EM is valuable, this is not the only prerequisite for having well-behaved high-quality usable grids for data collection. Additionally, for SPA data collection, sample quality in terms of preferential orientation, aggregation,

concentration on the grid surface, and protein disruption[35] are important parameters for optimizing the grid, which seem to evade light microscopy detection. It appeared that extensive aggregation of protein or vesicles could be observed on the grid by the current light microscopy setup, most probable through thickness variations of the water surface. We did not extensively study this further, though we like to point out that LM techniques for investigation of protein aggregation are proposed[36,37], and the use of additional spectroscopic or fluorescence LM methods during sample preparation of thin films might be worth exploring.

In contrast to other recently developed specimen preparation devices[29] we did not try to minimize the amount of used sample for the application on the grid. Instead, here we opted for developing a device with broad applicability and therefore evaluated its usability for a variety of different samples. Apoferritin was used as a SPA workflow standard to assess the resolution of the reconstruction. Using a single vitrification session we were able to prepare 9 grids of three samples, of which we used one, without any optimization for ice thickness, for an overnight data collection leading to a 2.4 Å resolution 3D map. We also prepared DNA origami cube samples for cryo-ET. This type of sample is known to easily aggregate and stick to air–water surfaces. Our experiments gave very similar results compared to experiments using other vitrification devices. In the same session we also prepared samples for cryo-EM on several types of liposomes, which showed that these samples can be easily prepared without

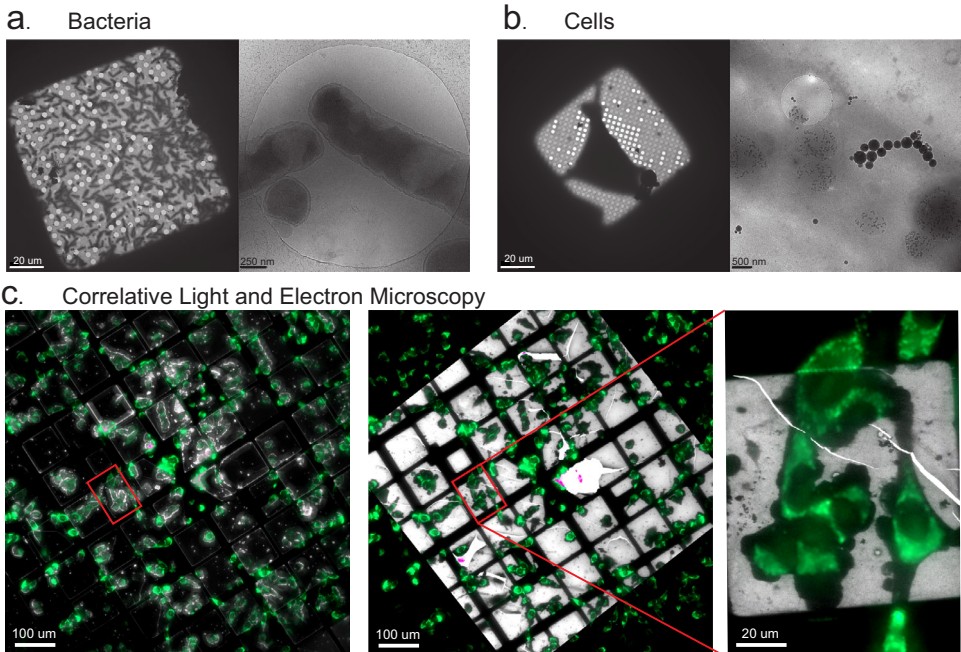

**Fig. 6 Application results, cells, and CLEM. a** Cryo-electron microscopy of bacteria. Typical cryo-EM overview of a single mesh square of a bacterial sample showing the ice thickness and distribution of bacteria (left) and a view within one hole in carbon film (right). **b** Typical cryo-EM overview of a single mesh square of cells grown on an EM grid (left) and cryo-EM image in a thin part of the cell showing internal vesicles and mitochondria with typical phosphate-rich electron-dense inclusions (right). **c** Cryo-CLEM of cells. Cryo light microscopy image overview of the central part of a finder grid which is an overlay of a dark-field image (white) and a fluorescence image (green) of fluorescently tagged cells (left), overlay of the cryoLM fluorescence image over the cryo-EM overview (white) (middle), red rectangle denotes the position of the mesh square shown at higher magnification (right).

much effort and optimization. More importantly, next to the particle suspension also bacterial and cellular cells could be prepared for both cryo-EM and cryo-CLEM imaging. The prepared samples and cryo-EM results were all indistinguishable from the same samples that were prepared using the Leica EM GP and Thermo Fisher Scientific Vitrobot Mark IV, despite the large differences in design and level of automation.

For bio-safety reasons, it is important to prevent the release of harmful substances. Also, it is important to avoid cross-contamination between samples. We noticed that without proper cleaning of the tweezer tips or exchange of the suction tubes, one can get cross-contamination of samples between grids that are subsequently produced. Cleaning and decontamination can be achieved by a combination of using exchangeable elements, rinsing, and high-temperature autoclaving measures. The sample-dipping containers as well as the sample pick-up cartridges are interchangeable and can be autoclaved or disposed of, all silicone tubing is disposable and can be interchanged easily. The line-up of the dipping baths is flexible and rinsing or cleaning steps for the tweezer tip can be configured in the software and automated. The tweezer assembly, the stainless suction tubing, and the mechanical modules in the direct vicinity of the sample during the suction process can easily be exchanged, or removed and autoclaved. The humidity chamber can be rinsed with, e.g., isopropyl alcohol, acetone, or ethanol and the design has large radii to facilitate cleaning. Although it was not part of the prototype tested in this paper we have added and tested HEPA-filters and heated high-temperature (~200 °C) exhaust channels to inactivate biologically active components in newer versions.

The device presented in this paper is highly automated and during preparation, user interaction is only needed to determine the desired moment for plunging. Future developments include the automatic detection of the optimal timepoint for vitrification. The experimental results reported here, indicate that the thin-film

interference fringes, that appear, change and disappear while the layer thickness is reduced, provide a direct and clear indication of the resulting film thickness after vitrification of the specimen, are easy to judge by the user. While in the current setup the decision when to plunge is taken by the user, we envision that in the future this decision will be implemented via a conventional or AI-based machine vision system. In addition, with spectrally resolved detection and analysis, the quantitative thickness of the liquid film could be determined optically, making the vitrification process completely user-independent.

## Methods

**Sample preparation.** Horse spleen apoferritin was purchased from Sigma (9013-31-4) and purified immediately before vitrification by gel filtration on a Superdex 200 Increase 3.2/300 column (Cytiva) in a 150 mM NaCl. 25 mM Tris pH 7.5, 2 mM DTT buffer. The peak fraction was collected and used for sample preparation at a final concentration of 2.5 μM.

Liposomes containing dimyristoylphosphatidylcholine (DMPC), dimyristoylphosphatidylglycerol (DMPG), cholesterol, and DNP-cap-PE (44:5:50:1 mol%) were prepared as previously described in 2019 by Lubbers et al. [38]. Lipids (Avanti Polar Lipids, AL, USA) were dissolved in chloroform–methanol (9:1 v/v) and dried under a nitrogen gas flow overnight. A final lipid concentration of 0.8 mg/ml was obtained, by rehydrating the lipid film in 20 mM sulforhodamine B (S1402; Sigma Aldrich, Missouri, USA) in phosphate-buffered saline (PBS), pH 7.4 at 37 °C for 30 min. The rehydrated lipid mixture was sonicated for 5 min at 37 °C to form liposomes, which were purified via size-exclusion chromatography using a prepacked NAP-25 column (17-0852-01; GE Healthcare, Little Chalfont, UK). Liposomes were mixed with 5 μg/ml IgG1 DNP monoclonal antibodies[39].

Liposomal amphotericin B (Ambisome®) was used in formulations as described earlier[40,41] after 10× dilution in water.

DNA origami cubes were designed using TALOS[42] using the standard cube presets with an 84-nucleotide edge length. Folding solutions were made in PCR tubes by mixing M13mp18 ssDNA (20 nM, Bayou Biolabs) with the appropriate ssDNA staples (200 nM of each staple, Integrated DNA Technologies; see Supplementary Data 1) supplemented with 20 mM MgCl₂ in a total volume of 50 μL. DNA origami cubes were thermally annealed in a Bio-Rad C1000 Touch™ Thermal Cycler using the following protocol: 80 °C down to 76 °C at a rate of 5 min/°C, 75 °C down to 30 °C at a rate of 13.75 min/0.5 °C, 29 °C down to 20 °C at

a rate of 10 min/°C. Ten 50 μL folding solutions were pooled together and subsequently purified and concentrated using Amicon® ultra 0.5 mL centrifugal filters (MWCO: 100 kDa).

17Clone1 mouse cells were prepared as described earlier[43] and fluorescently labelled for 1 h with MitoTracker™ (M7514, Invitrogen) at a final concentration of 0.5 μM from a 1 mM stock in DMSO. All samples were prepared on Quantifoil 300 mesh copper grids with R2/2 carbon film.

**Single particle acquisition**. Grids were loaded into a Titan Krios (Thermo Fisher Scientific) operated at 300 kV, equipped with a Gatan K3 BioQuantum direct electron detector. Movies with 50 frames and an accumulated dose of 65 e/Å² were acquired in counting mode using EPU software (Thermo Fisher Scientific) at a magnification of ×105,000, corresponding to a calibrated pixel size of 0.836 Å/pixel with a defocus range of −0.6 to −2.5 μm. A total of 2348 movies were collected over two microscopy sessions at The Netherlands Centre for Electron Nanoscopy (NeCEN). Detailed data acquisition parameters are summarized in Supplementary Fig. 1 and Supplementary Table 1.

**Single particle reconstruction**. RELION-3.1 beta software[44,45] was used for all image processing. Briefly, collected movies were subjected to beam-induced drift correction using MotionCor2[46], the contrast transfer function was estimated by CTFFIND-4.1.18[47]. RELION Gaussian picker was used to automatically pick 689,633 particles. After two rounds of 2D classification, false positives and contaminating features were discarded resulting in a 91,000 particles dataset. A previously determined apoferritin map filtered to 40 Å was used as a reference for 3D refinement. The final set of 91,000 particles was subjected to CTF refinement for optical and beam-tilt aberration corrections as well as per-particle defocus, per-micrograph astigmatism correction followed by Bayesian polishing[45,48]. A second 3D refinement was then performed yielding a 2.4 Å map. Map resolutions were estimated at the 0.143 criterion of the phase-randomization-corrected FSC curve calculated between two independently refined half-maps multiplied by a soft-edged solvent mask. Final reconstructions were sharpened and locally filtered in RELION post-processing. The X-ray model of horse spleen apoferritin (PDB entry 6RJH[49]) was fitted into the EM density by using the "fit in map" function within UCSF Chimera[50] version 1.13.1 after decreasing the EM map pixel size from 0.836 to 0.82 Å/pixel for a better fit. Maps were displayed using UCSF 1.13 and ChimeraX[51].

**Cryo-electron microscopy and tomography**. Cryo-EM images were recorded on an FEI Tecnai T12 Biotwin with LaB₆ source, operating at 120 keV on an FEI Eagle 4k × 4k CCD camera, using a Gatan 626 side entry cryo holder. For correlation with the latest recorded LM image before vitrification, low magnification (<×200) images were recorded by hand, covering the whole grid. Composite image overviews of the whole grid and overlays with the LM images were made in Adobe Photoshop.

Cryo-ET was performed on a Titan Krios (Thermo Fisher Scientific) operated at 300 kV, equipped with a Gatan K3 BioQuantum direct electron detector. Tilt series were recorded using TOMO 4 software (Thermo Fisher Scientific) between −56° to +56° with a tilt step of 2°, starting at 0°, with a total dose of 100 e/Å² and 16 frames per image at a nominal magnification of ×19.500, corresponding to a calibrated pixel size of 4.4 Å/pixel with a defocus of −7 μm. Cryo-electron tomography tilt series were reconstructed using IMOD software[52,53]. Surface renderings were performed by hand using Amira software (Thermo Fisher Scientific).

**Cryo correlative light and electron microscopy**. Cryo-LM was performed on a Zeiss Axioimager M2 equipped with a Linkam CMS196M. Transparent light and fluorescent image stacks (21 slices every 75 μm) we recorded using an EC Plan-Neofluor ×10/0.30 Ph1 objective lens on an AxioCam MR R3 at a pixel size at the sample level of 1 μm × 1 μm. Brightfield images were recorded using a LED light source and fluorescent images were recorded using an HXP 120 V light source using a 38 HE filter set. Maximum intensity projections for both image stacks were made using Zeiss ZEN 3.4 software and overlays were made in Adobe Photoshop 2021. Cryo-CLEM was performed on a Talos Arctica (Thermo Fisher Scientific) operated at 200 keV, equipped with a Gatan K3 BioQuantum direct electron detector. Correlations were performed and images were recorded using MAPS software (Thermo Fisher Scientific).

**Reporting summary**. Further information on research design is available in the Nature Research Reporting Summary linked to this article.

## Data availability

The apoferritin cryo-EM map generated in this study have been deposited in the EMDB under accession code EMD-13738. Video visualizations were created using Adobe Premiere Pro, Maxon Cinema 4D and Maxon Redshift and are available as Supplementary Movies 1 and 2. The data that support this study are available from the corresponding author upon request.

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

## Acknowledgements

We thank Meindert Lamers, Willem Noteborn, Leoni Abendstein, Georg Wolff (CCB, LUMC) for preparation of the samples. This work benefited from access to the Netherlands Centre for Electron Nanoscopy (NeCEN) at Leiden University, an Instruct-ERIC centre. The electron microscopy within this work is part of the research programme National Roadmap for Large-Scale Research Infrastructure (NEMI), project number 184.034.014, which is financed by the Dutch Research Council (NWO).

## Author contributions

R.I.K.: conceptualization, methodology, validation, investigation, writing—original draft, visualization, supervision. M.v.N.: conceptualization, methodology, validation, investigation. H.V.: conceptualization, methodology, validation, investigation. P.A.G.: Software. A.J.K.: writing—review and editing, funding acquisition. A.K.: conceptualization, methodology, supervision, funding acquisition. W.J.: SPA data collection. L.L.R.R.: data analysis. M.S.: conceptualization, methodology, writing—review and editing, supervision.

## Competing interests

Linkam Scientific Instruments Ltd. was granted European patent EP3018467 A1 "Microscopic Sample Preparation" (Inventors A.C.F.K., M.v.N., H.V., R.I.K., and M.S.); P.A.G. is employee of Linkam Scientific Instruments; A.C.F.K. is owner of Linkam Scientific Instruments Ltd., UK. The remaining authors declare no competing interests.
