## [Peer Review File · Nature Communications]

REVIEWER COMMENTS

Reviewer #1 (Remarks to the Author):

This is a well written manuscript of general interest to cryoEM practitioners where a novel device is proposed for sample vitrification.

The novel features of the instrument are dipping the grid in the sample and removing excess liquid through aspiration. This is very different than traditional methods that use filter paper to blot off excess sample.

The instrument is described in detail and several tests cases are shown.

Major Concerns

Assessment of film thickness. A camera is integrated to assess the film thickness to qualitatively assess the film thickness by examining the interference fringes to decide when to plunge the sample into liquid ethane. This is an old school assessment method that goes back to the days of manual plungers where film thickness would be visually assessed in the same way. My concerns are that this method may not be adequate to assess the thin films (< 200 or 100 nm) that are needed for high resolution cryoEM of small molecules. Some estimate of the theoretical limitations of this method should be made. For example, is the thickness proportional to $\lambda/2$? What about this case of white light illumination and can this even be determined with the transmitted and reflected light illumination? What fringes would be expected if the vitrified layer was less 200nm?

An additional concern is that there is no quantitative measurement of the ice thickness and correlating with what is seen in the videos. This may be beyond the scope of the paper but I don't think that claims can be made that an optimal moment of vitrification can be made without a quantitative analysis. Is the aspiration method capable of producing <200nm thick films?

Yield and reproducibility. There is no discussion of what the yield of suitable grids is. For example, 9 grids were made from the ApoF sample but only one was used in the data collection. Were the rest of the grids suitable for data collection or were only 1 out of 9 grids acceptable? The discussion states that 30-60% of the grid squares were usable but is that for one grid or based on assessment of

many grids? The ApoF resolution to 2.4Å from ~700,000 initial particles is not very impressive. Was this because the ice film thickness was too thick?

Minor Concerns

- Detailed plunger description.
- Should describe the amount of sample needed upfront in the description rather than later in the discussion. Can excess sample be recovered from the sample reservoir?

- Controlling Ice Thickness
- Reference to figure 2E,F should be 3E,F.
- It is not clear what is going on in supplementary movie 2. I understand that the authors are showing minimal evaporation in minutes 1-6 but the ice seems to get thicker (or at least the band of ice around the square seems to get broader in minutes 3-6). Is this water condensing from the air onto the grid? What are the authors trying to show with the additional 3 suction steps at minutes 6-9?

- Practical workflow.
- Should describe timing at each step. How long are typical glow discharge times and grid soaking times? How long is the aspiration stage?

- Other
- Would the instrument work with clipped grids?

Reviewer #2 (Remarks to the Author):

With their work, Koning and colleagues are addressing one of the most challenging practical hurdles in cryo-electron microscopy, namely, the preparation of cryo-samples. They have developed an innovative approach for sample removal from the support grid based on suction. In addition, the process is monitored in real-time with a camera, which allows control of the timing of the cryo-plunge for achieving a desired ice thickness. Their work will be of great interest to the rapidly growing community of researchers who use cryo-EM for their structural studies, and therefore I

strongly recommend its publication in Nature Communications. It would be nice if the authors could address the questions below in the revisions.

Pending questions:

1. A figure or a panel with a diagram of the arrangement of the suction tubes in relation to the grid would be very informative. From the current figures it is difficult to imagine what the airflow around the grid would be during suction.
2. Apart from the free water at the bottom of the environmental chamber, is there a plan to incorporate active humidity control or monitoring of the chamber?

Minor points:

- Reference 27 has been published "Cryo-EM structures from sub-nl volumes using pin-printing and jet vitrification", Nat. Commun. (2020)
- The manuscript would benefit from syntax/grammar proofing.

Radostin Danev

Reviewer #3 (Remarks to the Author):

Koning et al describe a fully automated procedure for cryoEM grid preparation. The solution described consists of a glow-discharging unit, sample delivery and suction (instead of blotting) to remove the excess fluid. All the movements and controls are robotic and monitorable, therefore the preparation is likely highly reproducible. The solution is neat and this reviewer finds it of interest to the community. On the other hand, innovation per se is lacking. The addition of robotics is useful but is not different from most industrial production processes with fixed positions. The use of a suction tube instead of blotting is nice, as it consents to control the sample removal and the speed at which this occurs but it does not solve the biggest problem presented by blotting, which is the segregation of molecules at the air-water interface.

I believe that this article is better suited to a specialised journal, where it would find an easy publication as the study is thorough and the manuscript was extremely well written.

REVIEWER COMMENTS

Reviewer #1 (Remarks to the Author):

This is a well written manuscript of general interest to cryoEM practitioners where a novel device is proposed for sample vitrification.

The novel features of the instrument are dipping the grid in the sample and removing excess liquid through aspiration. This is very different than traditional methods that use filter paper to blot off excess sample.

The instrument is described in detail and several test cases are shown.

Major Concerns

Assessment of film thickness. A camera is integrated to assess the film thickness to qualitatively assess the film thickness by examining the interference fringes to decide when to plunge the sample into liquid ethane. This is an old school assessment method that goes back to the days of manual plungers where film thickness would be visually assessed in the same way. My concerns are that this method may not be adequate to assess the thin films (< 200 or 100 nm) that are needed for high resolution cryoEM of small molecules. Some estimate of the theoretical limitations of this method should be made. For example, is the thickness proportional to $\lambda/2$? What about this case of white light illumination and can this even be determined with the transmitted and reflected light illumination? What fringes would be expected if the vitrified layer was less 200nm?

Reviewer #1 is correct that the interference fringes method we use for film thickness assessment is indeed not new and, as we have used and described it here, it does not adequately assess thin film thicknesses thinner than 200 nm if the film is illuminated with white light (visible part of spectrum) and is observed by eye or color-camera. Though we are aware that especially film thickness estimation below 200 nm is very relevant for high-resolution cryo-EM, as stated, this is out of the scope of the current manuscript. As far as we know no other known plunging setup is currently able to determine the film thickness accurately and quantitatively, though this clearly is a desirable feature. With the current setup the user can observe the film thinning, e.g. the progression through the sequence of interference colors associated with the thinning of the liquid layer.

To comment more specifically on the thin film interference: the coloring and interference depend on the thickness and refractive index of the film, in our case mostly water with refractive index close to 1.33. The situation is comparable to that observed for soap bubbles. A phase change of 180 degrees happens at the front of the film (first air-water interface), reflected intensity approaches zero for zero thickness and the fringe amplitude decreases with increasing order number and layer thickness.

[redacted]

[redacted]

[redacted]

Though we clearly observe fringes it is difficult to determine the exact color and fringe order in our setup and correlate the fringes with the originating thickness. In practice however, the resulting layer thicknesses in cryo-EM are estimated between ~50 and ~200 nm (see below), the user can observe the interference coloring down to a thickness between 100 and 200 nm, but not very accurately. Until then, cycling through the interference colors will at least indicate while the film is known to be too thick, and in practice gives usable results.

In our view it is possible to exploit the principle of thin-film interference to accurately and quickly determine water layer thickness. The outlook of a proper spectral analysis of the thin film interference spectrum ideally combines current visible light wavelengths and shorter wavelengths, towards the UV.

It is clear that the manuscript benefits from a clear description of the methods current limits and a discussion on what the possibilities are. Therefore we have made small changes in the manuscript concerning the claims and added the following paragraph on thin film measurement to the discussion: “

*“Though white light thin layer interference can give detailed information on film thickness (Afanasyev YD, Andrews GT, Deacon CG. Measuring soap bubble thickness with color matching. American Journal of Physics **79**, 1079-1082 (2011).), in the current setup with transmission and reflected light, the exact color cannot be determined. Also, below ~100 nm thin films are colorless and only changes in intensity, which cannot be related to a specific thickness since these interfere with background intensity differences. However, quantitative thin film analysis in the range of 50 to 200 nm is important for determining optimal thickness for high-resolution cryo-EM of small particles and it would be a desirable feature for a plunger setup. With appropriate spectrally resolved detection and analysis methods the quantitative thickness of the liquid film could be determined optically. In principle, interference color analysis using light of three wavelengths (Kitagawa K. Thin-film thickness profile measurement by three-wavelength interference color analysis. Applied optics **52**, 1998-2007 (2013).) or a combination of holography and interferometry (Ferraro V, Wang Z, Miccio L, Maffettone*

PL. Full-Field and Quantitative Analysis of a Thin Liquid Film at the Nanoscale by Combining Digital Holography and White Light Interferometry. The Journal of Physical Chemistry C 125, 1075-1086 (2021) were demonstrated for full-field optical thin film measurement. Though measuring films below 100 nm will likely need shorter wavelengths.”

An additional concern is that there is no quantitative measurement of the ice thickness and correlating with what is seen in the videos. This may be beyond the scope of the paper but I don't think that claims can be made that an optimal moment of vitrification can be made without a quantitative analysis. Is the aspiration method capable of producing <200nm thick films?

We are aware that we did not do a thorough quantitative assessment of the water film thickness using light microscopy, nor did we perform a comparative quantitative analysis of the resulting vitrified layer using e.g. film thickness measurements with Energy Filtered Transmission Electron Microscopy. This indeed would be a very interesting topic and we might even follow up on that in the future, but this is out of the scope of the current manuscript. We can make quantitative estimate for the range of the film thickness because the maximum penetration depth for electrons through ice is known for a given electron acceleration voltage. We are aware that we have to be careful with the claims we made and we have carefully checked the manuscript for overinterpretations but did not find any specific unsupported claims. We do not agree with reviewer #1 that no claims at all can be made by correlating the last movie frames with the resulting quality of the grid - although it should be added that the sequence of interference colors and fringe depth during the thinning process gives additional information not captured in the last image.

For one, we think we can claim that a usable film thickness is reproducibly attainable with the current setup. First, we showed that for all types of samples, usable cryo-EM images can be acquired from every single produced grid. Though we did not specifically mention it, film thickness can be estimated from the sizes of the containing particles. In the images of the ferritin we predominantly observed non-overlapping particles, which suggests that the layer thickness is thicker than 1 particle to 2 particles thick (~60 - 120 nm) but not much larger than ~180 nm (3 particles thick). From the DNA origami structures (Figure 5B) similar thicknesses are estimated. Center parts of the hole contain 1 particle thick (~50 nm) while on the sides 2 to 3 particles can be seen on top of each other (~100-150 nm). These thicknesses also follow from cryo-electron tomography reconstructions. Furthermore the cryo-EM images from the lipid vesicles (Figure 5C) show that vesicles ranging from 50 in the center to 200 nm on the outside (possibly a bit flattened) embedded in the ice, again indicating a layer thickness range of 50 to 150 nm. In conclusion we are confident that the aspiration method can generate film layer thicknesses ranging from 50-150 nm, which apparently is the resulting vitrification thickness after the observed LM interference patterns, which are limited to approximately 200 nm.

To clarify the thickness we have added the following sentence to the manuscript discussion:

“Though the resulting thickness of the vitrified water layers was not quantitatively measured, on the basis of the particle size in the different samples we estimated that the resulting vitrified water layers vary between 50 and 200 nm in size.”

Yield and reproducibility. There is no discussion of what the yield of suitable grids is. For example, 9

grids were made from the ApoF sample but only one was used in the data collection. Were the rest of the grids suitable for data collection or were only 1 out of 9 grids acceptable? The discussion states that 30-60% of the grid squares were usable but is that for one grid or based on assessment of many grids?

We did not elaborate on the yield of the suitability of grid, which indeed we should do. In practice we have counted the percentage of squares that had an appearance as in Figure 3 E and F against the total amount of squares that appears on a whole grid atlas. We did this for 16 grids with three different single particle samples. We did not include the bacterial and cellular samples as these did not show a strong variation in water thickness since bacteria and cells hold water and the usability depends more on the sample itself than the blotting/aspiration process. The usability ranged from an average 30% (conservative estimation, all carbon foil holes within a grid square are usable) to 60% (optimistic estimation, not all carbon foil holes within a grid square are usable). The variation between the grids was not high, all 16 grids in the quantification were usable for cryo-EM. Furthermore, thickness variation was present in each grid, which resulted in enough holes for SPA data collection in every single grid.

We added a part in the discussion on the yield and reproducibility:

“Since visual inspection gives prior knowledge on the quality and usability of the grids for cryo-EM, only good grids are transferred into the TEM and used for data acquisition and we have a perfect yield of grids, while discarding the occasional grids that did not behave well during thinning, e.g. because of bending or broken support film, before plunging. For purified protein and liposomal samples, within a single grid typically one third of the squares (especially around the position of the suction tubes) is too dry and unusable for data collection, one third is usable (vast majority of foil holes are covered with vitreous water) and one third has roughly equal amounts of empty and filled foil holes, making the usable amount of squares per grid vary between 30% and 60% (n=16). For grids with bacteria and adherent cells the yield was much better since water is better retained around these structures and they are less sensitive for drying.”

The ApoF resolution to 2.4Å from ~700,000 initial particles is not very impressive. Was this because the ice film thickness was too thick?

The final resolution in cryo-EM SPA, apart from ice thickness, depends on many factors. The apo-Ferritin reconstruction was calculated from randomly chosen positions on a grid of which the ice thickness might not have been optimal. We did not aim for the best statistics, since using a minimal amount of particles for the highest resolution was not our goal, we just wanted to make sure that by using aspiration we can get a suitable -not an optimal- thickness and a reconstruction on the first trial without any problems. Main problem was that the apo-Ferritin had a tendency to aggregate and we needed 3 preparations in order to reduce aggregation. An advantage was that this aggregating behavior was already visible by LM before blotting by thickness differences on the grid.

To clarify the thickness we have changed the following sentence in the discussion of the manuscript:

“Using a single vitrification session we were able to prepare 9 grids of which we used one for an overnight data collection leading to a 2.4 Å resolution 3D map.”

to

“Using a single vitrification session we were able to prepare 9 grids of three samples, of which we used one, without any optimization for ice thickness, for an overnight data collection leading to a 2.4 Å resolution 3D map.”

Minor Concerns

- Detailed plunger description.
- Should describe the amount of sample needed upfront in the description rather than later in the discussion.

We have moved the information on the amount of sample that is needed for application to earlier in the manuscript, from the discussion to the practical workflow part.

- Can excess sample be recovered from the sample reservoir?

Yes. Excess sample can easily be recovered from the reservoir.

- Controlling Ice Thickness
- Reference to figure 2E,F should be 3E,F.

We have corrected this error.

- It is not clear what is going on in supplementary movie 2. I understand that the authors are showing minimal evaporation in minutes 1-6 but the ice seems to get thicker (or at least the band of ice around the square seems to get broader in minutes 3-6). Is this water condensing from the air onto the grid?

In supplementary movie 2 on the right side it is noted in text what is going on: suction, water relocation, slow water sublimation.

What are the authors trying to show with the additional 3 suction steps at minutes 6-9?

Here we show that we can control thickness, but indeed this strictly is not necessary.

- Practical workflow.
- Should describe timing at each step. How long are typical glow discharge times and grid soaking times? How long is the aspiration stage?

Good point, we added the amount of time needed for all steps.

We should note that timing comparison of the whole grid preparation process with classical methods is more difficult since these also include startup times of used (glow discharge and vitrification) apparatuses, and this also depends on the type of instruments and the setup in the labs. Though we did not specifically measure it, we estimate that for the complete process (of making e.g. 12 grids) using the Linkam plunger is in total about 2x faster than a Leica EM GP or TFS Vitrobot. We have no knowledge on the speeds of the Vitrojet and Chameleon. Biggest time savers

are the reproducibility and foreknowledge of the grid quality.

- Other

- Would the instrument work with clipped grids?

The instrument does not currently work with clipped grids. Though the gripping mechanism can be adapted for clipped grids and the suction will work with clipped grids, especially plunging of clipped grids into liquid ethane will not result in a fast enough freezing rate to get vitrified water. I understand that this is one reason for the use of jets of liquid ethane for vitrification in the Vitrojet (See Reference 27 in the manuscript). This is not possible in our small setup.

Reviewer #2 (Remarks to the Author):

With their work, Koning and colleagues are addressing one of the most challenging practical hurdles in cryo-electron microscopy, namely, the preparation of cryo-samples. They have developed an innovative approach for sample removal from the support grid based on suction. In addition, the process is monitored in real-time with a camera, which allows control of the timing of the cryo-plunge for achieving a desired ice thickness. Their work will be of great interest to the rapidly growing community of researchers who use cryo-EM for their structural studies, and therefore I strongly recommend its publication in Nature Communications. It would be nice if the authors could address the questions below in the revisions.

Pending questions:

1. A figure or a panel with a diagram of the arrangement of the suction tubes in relation to the grid would be very informative. From the current figures it is difficult to imagine what the airflow around the grid would be during suction.

Indeed this was not clear from any figure. We have added the positions of the suction tubes that were used in the described experiments in Figure 4 (and the figure legend were adapted accordingly). The tubes are positioned on the sides of the grid, touching both the outside ring as well as part of the grid squares. It must be noted that the suction tube is modular and can easily be interchanged in order to alter the number of tubes and their positioning.

The 3D arrangement of the suction tubes is within the “tweezer lock”, which is such that the airflow is rather contained.

2. Apart from the free water at the bottom of the environmental chamber, is there a plan to incorporate active humidity control or monitoring of the chamber?

Currently there are no plans to incorporate active humidity control. We hope to have showed here that this is not strictly necessary. Our device measures humidity, temperature and substrate temperature making it possible to obtain dewpoint information. Currently there are no plans to add an active ultrasonic nebulizer or similar. Space in the current humidity chamber is also quite restricted. By creating a dewpoint controlled mini environment in the temperature controlled “tweezer lock” we have control over the evaporation from the grid. Without active humidity control the monitoring of all temperatures is of vital importance and this is what we incorporated, and possibly expand when necessary.

Minor points:

- Reference 27 has been published “Cryo-EM structures from sub-nl volumes using pin-printing and jet vitrification”, Nat. Commun. (2020)

We have updated the reference.

- The manuscript would benefit from syntax/grammar proofing.

We use the program “Grammarly” (<https://www.grammarly.com/>) to check the grammar and checked over 200 issues.

Reviewer #3 (Remarks to the Author):

Koning et al describe a fully automated procedure for cryoEM grid preparation. The solution described consists of a glow-discharging unit, sample delivery and suction (instead of blotting) to remove the excess fluid. All the movements and controls are robotic and monitorable, therefore the preparation is likely highly reproducible. The solution is neat and this reviewer finds it of interest to the community. On the other hand, innovation per se is lacking. The addition of robotics is useful but is not different from most industrial production processes with fixed positions. The use of a suction tube instead of blotting is nice, as it consents to control the sample removal and the speed at which this occurs but it does not solve the biggest problem presented by blotting, which is the segregation of molecules at the air-water interface. I believe that this article is better suited to a specialised journal, where it would find an easy publication as the study is thorough and the manuscript was extremely well written.

Reviewer #3 is correct that the current big problem of cryo sample preparation for single particle reconstruction techniques is the preferential positioning of proteins at the air water interface, and its resulting preferential orientation and even denaturation. We think other measures than a new plunging device are necessary to overcome these problems and indeed the newly designed plunger described in this manuscript was not intended to solve these issues.

Our focus was on a broad usability, including bacterial and cellular samples and tomography and CLEM. Therefore we focused on more general specimen preparation problems, like thickness

control, reproducibility and speed. Also there is a growing user base that is helped by ease of use through automation. Though the individual steps might not be new as such, the concept as a whole is innovative – we have acquired a patent – and interesting to the community – as reviewer noted.

Additionally, we hope to have made clear in the manuscript that the vitrification device is broadly usable and deals with a broad variety of problems. Therefore, we also think it will be useful for the broad community that Nature Communication covers. Furthermore, we reckon that with the publication in Nature Communications of a similar vitrification device (Ravelli et al. 2020; Reference 27 in the manuscript), with comparable levels of automation and innovation, and maybe even with a less broad technical applicability, we hope our manuscript would be within the scope and relevant for the readers of Nature Communications.

REVIEWERS' COMMENTS

Reviewer #1 (Remarks to the Author):

The authors have addressed my concerns.